# Development of Antibodies against HPV-6 and HPV-11 for the Study of Laryngeal Papilloma

**DOI:** 10.3390/v13102024

**Published:** 2021-10-07

**Authors:** Taro Ikegami, Norimoto Kise, Hidetoshi Kinjyo, Shunsuke Kondo, Mikio Suzuki, Narutoshi Tsukahara, Akikazu Murakami, Asanori Kiyuna, Shinya Agena, Katsunori Tanaka, Narumi Hasegawa, Junko Kawakami, Akira Ganaha, Hiroyuki Maeda, Hitoshi Hirakawa

**Affiliations:** 1Department of Otorhinolaryngology, Head and Neck Surgery, Graduate School of Medicine, University of the Ryukyus, Okinawa 903-0215, Japan; ikegami@med.u-ryukyu.ac.jp (T.I.); norimoto7@gmail.com (N.K.); hidechanman223@yahoo.co.jp (H.K.); kouhouiinn@yahoo.co.jp (S.K.); jibika_asanori97@yahoo.co.jp (A.K.); harugen3@yahoo.co.jp (S.A.); mizuki0415@gmail.com (K.T.); br101426@gmail.com (N.H.); sscm.11om2mf@gmail.com (J.K.); maeidahiroyuki@yahoo.co.jp (H.M.); hanntagawa@hotmail.com (H.H.); 2Department of Parasitology & Immunopathoetiology, Graduate School of Medicine, University of the Ryukyus, Okinawa 903-0215, Japan; tsukahara@rephagen.com (N.T.); akimu@tokushima-u.ac.jp (A.M.); 3RePHAGEN Co., Ltd., Okinawa 904-2234, Japan; 4Department of Oral Microbiology, Graduate School of Biomedical Sciences, Tokushima University, Tokushima 770-8504, Japan; 5Department of Otolaryngology, Faculty of Medicine, University of Miyazaki, 5200 Kihara, Miyazaki 889-1692, Japan; ganaha.akira.t8@cc.miyazaki-u.ac.jp

**Keywords:** laryngeal papilloma, human papillomavirus 6, human papillomavirus 11, viral DNA and mRNA expression, anti-E4 antibody

## Abstract

Laryngeal papilloma (LP), which is associated with infection by human papillomavirus (HPV)-6 or -11, displays aggressive growth. The precise molecular mechanism underlying the tumorigenesis of LP has yet to be uncovered. Building on our earlier research into HPV-6, in this study, the viral gene expression of HPV-11 was investigated by quantitative PCR and DNA/RNA in situ hybridization. Additionally, newly developed antibodies against the E4 protein of HPV-6 and HPV-11 were evaluated by immunohistochemistry. The average viral load of HPV-11 in LP was 1.95 ± 0.66 × 10^5^ copies/ng DNA, and 88% of HPV mRNA expression was found to be *E4*, *E5a*, and *E5b* mRNAs. According to RNA in situ hybridization, *E4* and *E5b* mRNAs were expressed from the middle to upper part of the epithelium. E4 immunohistochemistry revealed a wide positive reaction in the upper cell layer in line with *E4* mRNA expression. Other head and neck lesions with HPV-11 infection also showed a positive reaction in E4 immunohistochemistry. The distribution pattern of HPV DNA, viral mRNA, and E4 protein in LP with HPV-11 infection was quite similar to that of HPV-6. Therefore, it might be possible to apply these E4-specific antibodies in other functional studies as well as clinical applications, including targeted molecular therapies in patients with HPV-6 and HPV-11 infection.

## 1. Introduction

Human papillomaviruses (HPVs), a family of viruses with double-stranded circular DNA, are responsible for various tumorous lesions, ranging from warts to cancers of the cervix, anus, and head and neck [1]. In particular, laryngeal papilloma (LP) is a benign tumor that has been clearly linked to infection with HPV-6 and -11. The incidence of LP is estimated to be around 4 per 100,000 children and about half that in adults [2]. Its clinical characteristics include a high recurrence rate, aggressive growth, and occasional tracheobronchial extension, especially in children [3]. Although malignant transformation of LP occurs in less than 3% of cases [4,5], multiple surgeries, including tracheostomy, may be required in order to maintain the airway [3]. Human papillomavirus oral papilloma is often sexually transmitted, but non-sexual modes of transmission should be considered [6], including hand-to-mouth transmission and vertical transmission during pregnancy or childbirth. According to a recent report [7], juvenile-onset LP is caused by only one type of HPV in the vast majority of cases (95.6%), consisting primarily of HPV-6 (79.6%) and to a lesser degree HPV-11 (15.4%). However, children with HPV-11 infection tend to have more severe disease, including greater numbers of papilloma lesions and the need for surgeries [7]. The reasons for the differences in clinical features between HPV-6 and -11 LP are not known.

The approximately 8-kbp HPV genome is divided into three sections: six early genes (*E1*, *E2*, *E4*, *E5*, *E6*, and *E7*), two late genes (*L1* and *L2*), and the long control region (LCR). HPV-6 and -11 detected in LP are categorized as low-risk HPV according to their association with cancer. HPV initially infects basal cells in the mucosal and cutaneous epithelium through membrane disruption [8]. Despite significant variations in open reading frames (ORF), the well-conserved core genes of HPV are observed in both the replication genes (*E1* and *E2*) and the packaging genes (*L1* and *L2*). The rest of the genes (*E6*, *E7*, *E5*, and *E4*) are highly diverse and their involvement is observed in the cell cycle, escape from immune surveillance, and viral release [9]. Although the role of viral gene expression in high-risk HPV has been comprehensively elucidated [8,9], there are few reports on the role of gene expression in low-risk HPV as it relates to LP. Compared with HPV-6, HPV-11 poses a higher risk for the development of LP, although both are alphapapillomaviruses with similar nucleotide sequences [10].

Previously, we developed a novel real-time PCR method to measure the amounts of the abovementioned early and late mRNAs as well as an mRNA in situ hybridization (ISH) assay for most of the early mRNAs. We also established a monoclonal antibody to immunohistochemically localize E4 in HPV-6 LP [11]. The results revealed that 96% of the expression of nine viral mRNAs were due to expression of *E4*, *E5a*, and *E5b* mRNAs. Moreover, the viral DNA load change during recurrence of LP was similar to the change in mRNA expression. These results suggest that *E4*, *E5a*, and *E5b* genes may be expressed in a coordinated manner for the purposes of viral replication, release, and elusion from host immune surveillance in HPV-6. The expression of these genes was observed in the epithelium along with the dynamic alteration of HPV viral load in both HPV-6 infected primary tumors and recurrent tumors after treatment [12].

In the present study, given that HPV-11 causes more severe disease than HPV-6 [7], we further developed our real-time PCR assay, mRNA-ISH method, and anti-E4 antibody to facilitate molecular and immunohistochemical analyses of HPV-11 LP. Because there are no available antibodies that do not cross-react with HPV-6 and HPV-11 in immunohistochemical examinations, the sensitivity and specificity of our anti-E4 antibody against HPV-6 and HPV-11 were validated using various samples from the head and neck region, including inflammation, benign lesions, and cancerous lesions.

## 2. Materials and Methods

### 2.1. Subjects

Specimens were collected from the following patients at our hospital between 2005 and 2021: 28 patients with LP, 10 with laryngeal cancer, 5 who underwent tumor-free vocal cord removal during pharyngolaryngoesophagectomy, 5 with chronic tonsillitis, 5 with oropharyngeal cancer, and 5 with tongue cancer, 4 with nasoseptal exophytic papilloma, and 1 each with paranasal exophytic papilloma, inverted papilloma with squamous cell carcinoma, nasopharyngeal papilloma, and oropharyngeal squamous papilloma. The HPV genotypes of the oropharyngeal and laryngeal cancer samples were analyzed in our previous studies [13,14]. All specimens were frozen in liquid nitrogen immediately after biopsy or surgical excision and stored at −80 °C until analysis.

### 2.2. Detection of HPV DNA

DNA was extracted from fresh-frozen samples and subjected to PCR using the degenerated consensus primer sets that were designed to amplify the L1 region, as in our previous studies [11,15]. We examined the presence and integrity of DNA in all samples, and HPV subtypes were determined. For further details, see the Appendix A.

### 2.3. Measurement of Viral DNA Load and mRNA Expression in HPV-11-Infected Papilloma by Quantitative Real-Time PCR

Total RNA was extracted from fresh-frozen papilloma samples and subjected to quantitative real-time PCR to measure the absolute levels of *E6*, *E7*, *E1*, *E2*, *E4*, *E5a*, *E5b*, *L2*, and *L1* mRNAs, as described in our previous studies [11]. More precisely, three clones (clones A, B, and C; Figure 1) were prepared using genomic DNA and used as standards for quantification. The primers and amplification efficiency of target genes are shown in Appendix A. Viral load was defined by *E6* copy number/ng cellular DNA. Details of the methods are provided in the Appendix A.

### 2.4. ISH with HPV DNA Probes

Biotinylated DNA probes were prepared and used to perform ISH of HPV DNA, as described in our previous study [11] and the Appendix A.

### 2.5. RNA-ISH with HPV-11 E6, E2, E4, and E5b Digoxigenin RNA Probes

The *E6*, *E2*, *E4*, and *E5b* genes of HPV-11 (Figure 1) were amplified to prepare digoxigenin RNA probes for RNA-ISH, as in our previous study [11]. RNA-ISH was performed as described previously [11], and further details are provided in the Appendix A.

### 2.6. Generation of an Anti-HPV-11 E1^E4 Polyclonal Antibody

#### 2.6.1. Preparation of the Target Antigen

To produce recombinant HPV-11 E1^E4 protein in *E. coli* as a target antigen, the entire *E1^E4* gene of HPV-11 was amplified with HPV-11 E1^E4-*Nde*I-F and HPV-11 E1^E4-*Xho*I-R or HPV-11 E1^E4-*Bgl*II-F and HPV-11 E1^E4-*Xho*I-R primers by the following PCR method. The primers are listed in Appendix A. The 12.5-μL PCR reaction mixture included cDNA of Patient 1 (1 μL), forward and reverse primers (0.24 μM), and GoTaq^®^ Green Master Mix (6.3 μL; Promega, Madison, WI, USA). PCR was performed under the following conditions: 95 °C for 5 min, followed by 35 cycles at 95 °C for 30 s, 60 °C for 30 s, and 72 °C for 1 min, and finally 72 °C for 5 min. The PCR product was amplified with HPV-11 E1^E4-*Nde*I-F and HPV-11 E1^E4-*Bam*HI-*Xho*I-R primers was digested with *Nde*I and *Xho*I, and then subcloned into the *Nde*I/*Xho*I restriction sites of the pET22b (+) vector (Merck KGaA, Darmstadt, Germany), resulting in the pET22b (+) 1× HPV-11 E1^E4 vector. Then, the PCR fragment amplified with the HPV-11 E1^E4-*Bgl*II-F and HPV-11 E1^E4-*Bam*HI-*Xho*I-R primers was digested with *Bgl*II and *Xho*I and subcloned into the *Bam*HI and *Xho*I restriction sites of the pET22b (+) 1× HPV-11 E1^E4 vector, resulting in the pET22b (+) 2× HPV-11 E1^E4 vector. Finally, the above PCR fragment digested with *Bgl*II and *Xho*I was subcloned again into the *Bam*HI and *Xho*I restriction sites of the pET22b (+) 2× HPV-11 E1^E4 vector, resulting in the pET22b (+) 3× HPV-11 E1^E4 vector. As the pET22b (+) vector contains a His-tag (6 histidine residues) at the C-terminal region of the multiple-cloning site, the pET22b (+) 3× HPV-11 E1^E4 vector was expected to express a 31.8-kDa protein (three tandem fused HPV-11 E1^E4 plus the His-tag). The pET22b (+) 3×-HPV-11 E1^E4 vector was transformed into *E. coli* BL21 (DE3). The transformed *E. coli* were cultured for 15 h at 25 °C in Luria Broth medium containing 100 mg/mL ampicillin and 1 mM isopropyl β-D-1-thiogalactopyranoside. After centrifugation at 20,000× *g* for 5 min, the pellet was suspended in ice-cold PBS and sonicated for 30 min. After centrifugation at 20,000× *g* for 10 min, the pellet was suspended in ice-cold PBS, including 6 M guanidine hydrochloride, and sonicated for 30 min. Then, dithiothreitol was added to the lysate (final concentration of dithiothreitol, 1 mM), mixed, and placed on ice for 1 h. After centrifugation at 20,000× *g* for 5 min, the supernatant was collected and purified with a His-trap column as described previously [16]. To remove guanidine hydrochloride dissolved in the protein solution, 0.5 mL of 3× HPV-11 E1^E4 protein solution (2.0 mg/mL) was dialyzed twice using a miniature dialysis machine with a molecular weight cutoff of 10,000 and a 0.1-mL conical tube (Slide-A-Lyzer; Thermo Fisher Scientific, Waltham, MA, USA) in 300 mL of 8 M urea solution for 2 h. Then, it was dialyzed twice in 300 mL of 4 M urea for 2 h. Finally, the 3× HPV-11 E1^E4 protein solution was replaced from 6 M guanidine hydrochloride with 4 M urea. The concentration of the obtained 3× HPV-11 E1^E4 protein solution was quantified based on the absorption measurement value at a wavelength of 280 nm.

#### 2.6.2. Preparation of the Polyclonal Antibody against HPV-11 E1^E4

A specific pathogen-free rabbit was immunized with the above 3× HPV-11 E1^E4 protein (Sigma-Genosys, Inc., Hokkaido, Japan). On day 0, initial immunization was administered with 400 μg of the protein combined with Freund’s complete adjuvant. Subsequent immunization was administered with Freund’s incomplete adjuvant, including 200 μg of the protein, at 2-week intervals (three times total). Antiserum was obtained by bleeding the rabbit on day 49.

### 2.7. Western Blotting

Antibodies against HPV-11 E1^E4 and HPV-6 E1^E4 were evaluated by Western blotting, as in our previous study [11]. For further details, see the Appendix A.

### 2.8. Immunohistochemistry Using the Newly Developed Anti-HPV-11 E1^E4 Antibody and Anti-HPV-6 E1^E4 Antibody

FFPE samples from 66 patients were used to evaluate the sensitivity and specificity of the newly developed anti-HPV-11 E1^E4 antibody and anti-HPV-6 E1^E4 antibody in immunohistochemistry, using the same process as in our previous study (for details, see Appendix A) [11]. The details of the samples analyzed are shown in Table 1

### 2.9. Statistical Analysis

Viral mRNA levels in terms of the percentage of viral mRNA expression were analyzed by the Kruskal–Wallis test. Statistically significant differences in the mRNA levels of the nine viral genes were detected using the Dwass–Steel–Critchlow–Fligner multiple comparison test. The correlations between viral load and viral mRNA levels were analyzed by Spearman’s rank-ordered test. Comparisons of viral load between positive and negative immunohistochemistry staining of HPV-6-infected papilloma (LP and nasoseptal exophytic papilloma) with the anti-HPV-6 E1^E4 antibody were performed by the Mann–Whitney U-test. *p* < 0.05 was taken to indicate a statistically significant difference. All statistical analyses were performed using NCSS ver.12 (NCSS, LLC, Kaysville, UT, USA).

### 2.10. Ethics Approval and Informed Consent

This study was conducted with the approval of the Institutional Review Board of the University of Ryukyus (project identification code 35) and conformed to the principles of the Declaration of Helsinki and its later amendments. Written informed consent was obtained from all participants prior to sample collection.

## 3. Results

### 3.1. HPV-11 DNA Distribution, Subtypes, and Viral Load

The clinicopathological characteristics of the patients with HPV-11 infection are summarized in Table 2 (i.e., three with LP, one with nasoseptal exophytic papilloma, and one with nasopharyngeal papilloma). The Derkay score [17] of LP ranged from 2 to 7 (Table 2). The average viral load was 1.95 ± 0.66 × 10^5^ copies/ng DNA (median, 7.15 × 10^4^, Table 2) in 11 specimens obtained from these five patients. Of the three patients with LP, Patient 3 had multiple tumor lesions and received frequent surgeries in the bilateral true vocal cords, epiglottis, and posterior wall of the hypopharynx. Viral load in Patient 3 varied extensively at the different surgeries and tumor locations (Table 2). For example, viral load was much smaller in epiglottal tumors than in tumors in the true vocal cords and hypopharynx. Moreover, there was a 10-fold difference in viral load in the true vocal cords and hypopharynx among the surgeries.

HPV DNA-ISH produced positive signals, mainly in the nucleus, although a weak positive cytoplasmic reaction pattern was also observed. The positive signals were distributed throughout the upper and middle layers of cells (Figure 2).

### 3.2. Expression of HPV-11 mRNAs

#### 3.2.1. Expression Levels of the Nine Viral mRNAs

The expression levels of the nine HPV-11 virus mRNAs were ascertained in 10 samples from three patients with LP, one with nasoseptal papilloma, and one with nasopharyngeal papilloma (Table 2). As shown in Figure 3a, the expression levels (mean ± standard error) of *E4, E5a*, and *E5b* mRNA were much higher (*E4*, 7.841 ± 4.948; *E5a*, 2.857 ± 1.394; *E5b*, 6.285 ± 3.346) than those of *E6*, *E7*, *E1*, *E2*, *L2*, and *L1* (*E6*, 0.117 ± 0.035; *E7*, 0.504 ± 0.167; *E1*, 0.750 ± 0.566; *E2*, 1.278 ± 0.945; *L2*, 0.057 ± 0.030; *L1*, 0.032 ± 0.013). Indeed, *E4*, *E5a*, and *E5b* accounted for 88% of the total expression of the nine viral mRNAs (*E4*, 36.66 ± 4.01%; *E5a*, 16.17 ± 2.15%; *E5b*, 35.36 ± 4.07%, Figure 3b), while the remaining six genes accounted for only 12% (*E6*, 1.01 ± 0.17%; *E7*, 4.06 ± 0.74%; *E1*, 2.17 ± 0.51%; *E2*, 4.01 ± 0.60%; *L2*, 0.31 ± 0.12%; *L1*, 0.23 ± 0.09%). There was no significant correlation between viral DNA load and the mRNA levels of *E6, E7, E1, E2, E4, E5a, E5b*, *L2*, and *L1*.

#### 3.2.2. mRNA Expression in LP Detected by RNA-ISH

The results of RNA-ISH revealed that *E4* and *E5b* mRNAs were markedly expressed and located in the cytoplasm of the upper third cell layer of LP (Figure 2). Meanwhile, *E2* mRNA was expressed in the nucleus of cells in the intermediate to upper third cell layer. *E6* mRNAs were not detected by RNA-ISH.

### 3.3. Western Blot Analysis Using the Anti-HPV-11 E1^E4 Antibody

Western blot analysis was performed to validate the polyclonal antibody against the HPV-11 E1^E4 that we developed. The results showed that the anti-HPV-11 E1^E4 antibody bound to the 13-kDa HPV-11 E1^E4-3× FLAG fusion protein but did not bind to the corresponding HPV-6 fusion protein nor the human CMTM7-3× FLAG fusion protein or a mock protein (Figure 4). The anti-HPV-11 E1^E4 antibody (1:5000 dilution) detected 50 μg of cell lysate transfected with HPV-11 E1^E4 but not 10 μg (Figure 4).

### 3.4. Immunohistochemistry Using the Anti-HPV-11 and Anti-HPV-6 E1^E4 Antibodies

Immunohistochemistry with the anti-HPV-11 E1^E4 antibody produced a positive reaction in the middle to upper third cell layers, much like the expression of *E4* RNA-ISH in HPV-11-infected LP (Figure 2 and Figure 5a). In addition to LP, HPV-11 E1^E4-positive signals were observed in nasoseptal exophytic papilloma and nasopharyngeal papilloma (Figure 5b,c). However, non-specific signals were observed in some capillary blood vessels of HPV-11-infected nasopharyngeal papilloma (Figure 5b). In contrast, signals were not detected in LP, normal vocal cord, chronic tonsillitis, and cancerous lesions in the larynx, oropharynx, and tongue without HPV-11 infection (Table 1 and Figure 5).

Immunohistochemistry with the anti-HPV-6 E1^E4 antibody showed positive signals in 16 of 21 HPV-6-infected LP and nasoseptal exophytic papilloma samples (Table 1 and Figure 6). A significant difference in the viral load of HPV-6 was observed between patients with positive signals (mean viral load, 450,081 copies/ng DNA) and negative signals (mean viral load, 9816 copies/ng DNA) (*p* = 0.0312, Mann–Whitney U-test). Meanwhile, signals were not observed in LP, normal vocal cord, chronic tonsillitis, and cancerous lesions in the laryngeal, oropharyngeal, and tongue samples without HPV-6 (Table 1 and Figure 6).

The sensitivity and specificity of the anti-HPV-11 E1^E4 antibody were both 100%, whereas those of the anti-HPV-6 E1^E4 antibody were 76.2% and 100%, respectively (Table 3).

## 4. Discussion

In this study, we developed an anti-HPV-11 E4 polyclonal antibody and clarified the intraepithelial distribution of the E4 protein. Using the anti-HPV-11 E4 polyclonal antibody developed in the present study and the anti-HPV-6 E4 monoclonal antibody developed in our previous study [11] and immunohistochemical examination of LP, we revealed the intraepithelial distribution of HPV-11-related E4 protein in nasal, nasopharyngeal, or pharyngeal samples. Moreover, our E4 antibodies against HPV-11 and HPV-6 did not show cross-reactivity. In the clinical setting, the detection of HPV type in LP is essential for predicting disease severity and recurrence. These antibodies were demonstrated to be useful for determining the HPV type in LP.

In addition, we examined viral load, in situ localization of viral DNA, and viral gene expression, including that of the “core” genes directly associated with viral DNA replication (*E1*, *E2*, *L1*, and *L2*), as well as functional “accessory” genes (*E6*, *E7*, *E4*, *E5a*, and *E5b*) in HPV-11-infected LP [10,18]. DNA-ISH showed more intense signals in the upper and middle layers than in the basal layers, suggesting that viral replication was activated more in the differentiated upper layers. We found that *E4* was the most highly expressed mRNA, followed by *E5b* and *E5a* in LP, which stood in stark contrast to the much lower expression of *E6*, *E7*, *E1*, *E2*, *L2*, and *L1*. We further investigated the intraepithelial distribution of *E4* and *E5b* transcripts in LP, finding that *E4* and *E5b* mRNAs were expressed in similar superficial epithelial layers. The DNA and RNA levels of HPV-11 genes and immunolocalization of E4 protein in HPV-11 resembled those of HPV-6 [11]. The predominant expression of E4 and E5 is a prominent feature of LP with HPV-11 and HPV-6 infection [11]. Thus, the mRNA expression patterns, HPV-DNA distribution, and localization of E4 protein in HPV-11-infected LP were quite similar to HPV-6-infected LP, although there were a limited number of specimens with HPV-11 infection.

The E4 protein encoded by HPV-16 DNA appears to be able to bind keratin as well as reorganize the keratin network of the cell, indicating possible involvement in the release and transmission of the virus [19]. Meanwhile, E5, which is a three-pass transmembrane protein [20], activates the epidermal growth factor receptor, inhibits the expression of the p21 tumor suppressor gene [21,22], and plays a role in elusion from immunosurveillance [23]. Although the biological functions of E5a and E5b in low-risk HPV remain poorly understood, they are expected to be three-pass and one-pass transmembrane proteins, respectively [24]. According to the PapillomaVirus Episteme (PaVE) database (https://pave.niaid.nih.gov/#explore/transcript_maps/hpv11 (accessed on 21 September 2021)), the following three promoters have been identified in the HPV-11 genome: E6 promoter (nt 90), E7 promoter (nt 264), and E1 promoter (nts 674–714). The PaVE database also shows 19 polycistronic messages transcribed by these promoters, among which *E5a* and *E5b* are the most encoded genes (*n* = 18), followed by E4 (*n* = 15, including the full-length message of E2), E2 (*n* = 7), E7 (*n* = 4), E6 (*n* = 3), L1 (*n* = 2), and E1 and L2 (*n* = 1). The expression of each viral gene examined in this study was in good agreement with the number of genes encoded by the message (Figure 3b). According to the PaVE database, *E5a* and *E5b* are always transcribed simultaneously, but the level of *E5b* expression was markedly different from that of *E5a*, with *E5b* showing a similar level of expression as *E4* in this study. In our previous study, HPV-6 mRNA expression of *E5a* and *E5b* differed greatly in LP samples, consistent with the results of the present study. Although the precise reason for the observed differences in *E5a* and *E5b* mRNA expression is not clear, the splicing of *E5a* and *E5b* mRNAs might occur differently in different LPs. Further study is needed to clarify how mRNA splicing occurs in low-risk HPV.

The life cycle of the virus synchronizes with the program of epithelial differentiation, possibly through the action of differentially expressed cellular transcription factors binding to the LCR throughout multiple layers of epithelia [25]. Viral pathogenicity depends on a number of factors, including the viral genotype, the type of cell infected, and the status of host immunity. In general, HPVs propagate through virion production upon infecting the epithelium, and their life cycles are highly dependent on epithelial differentiation. Although the host immune system virtually excludes or suppresses the majority of viral infection and reproduction, a small number of infected cells undergo successful viral reproduction with atypical epithelial differentiation for tumorigenesis [18]. Previous reports suggested that regulatory mechanisms operating in the LCR might be essential for ascertaining the tissue selectivity of various types of HPV [26,27]. Completion of the viral life cycle thus necessitates immune evasion mechanisms. Patient 3, who carried HPV-11, had multiple lesions during many surgeries. At the 16th surgery, there was a sufficient difference in viral load among the sites (Table 2). These findings suggest that the location of HPV infection and local immunity are important for the persistence and aggressiveness of HPV infection. It is expected that prophylactic HPV vaccines targeting HPV-6 and -11 may reduce the burden of anogenital warts and LP [28]. Further study is needed to clarify the differences in the clinical features of both HPV types. The specific antibodies against the E4 proteins of HPV-11 and HPV-6 in the present study are thus applicable to further functional studies and potential clinical therapies such as targeted molecular therapies.

## Figures and Tables

**Figure 1 viruses-13-02024-f001:**
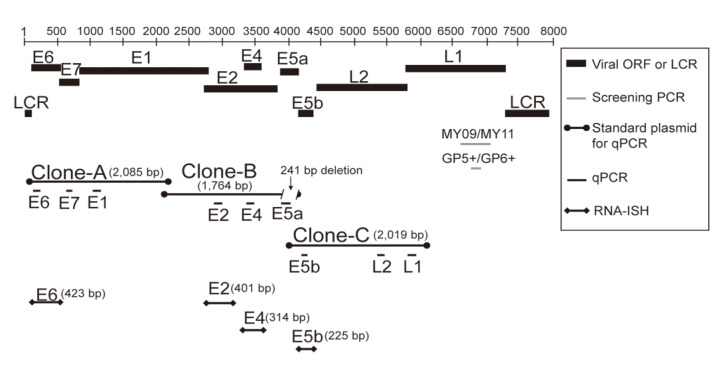
The HPV-11 genes, plasmid clones, and RNA-ISH probes used in this study.

**Figure 2 viruses-13-02024-f002:**
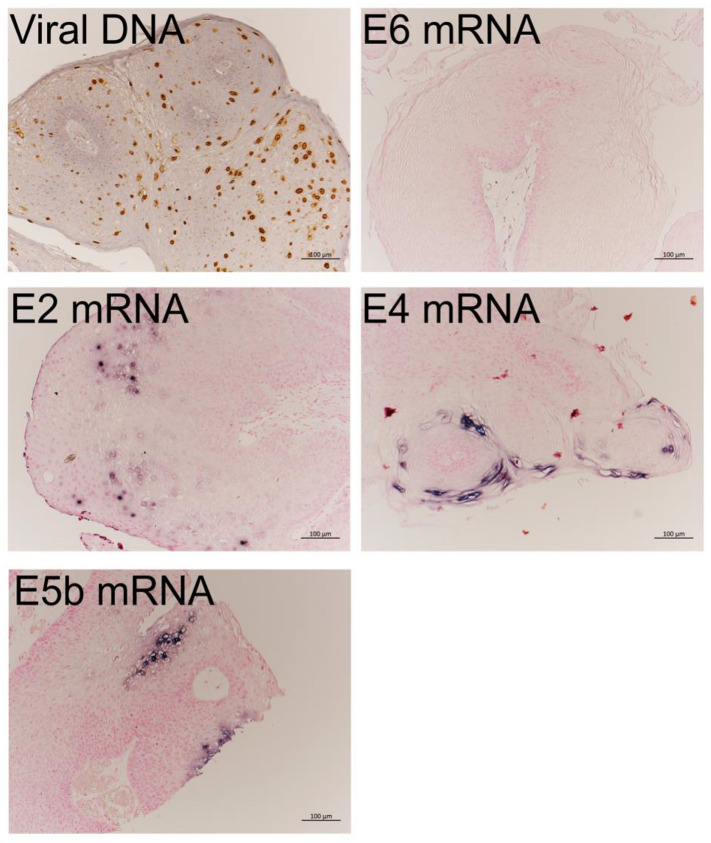
HPV-11-positive cells detected by DNA-ISH and RNA-ISH in LP. HPV-11 DNA was strongly observed from the middle to upper third cell layers in this representative case. A signal for *E6* mRNA was not observed. *E2* mRNA was weakly expressed from the middle to upper third layers. *E4* and *E5b* mRNAs were strongly expressed in the upper third cell layer in LP. Bar = 100 μm.

**Figure 3 viruses-13-02024-f003:**
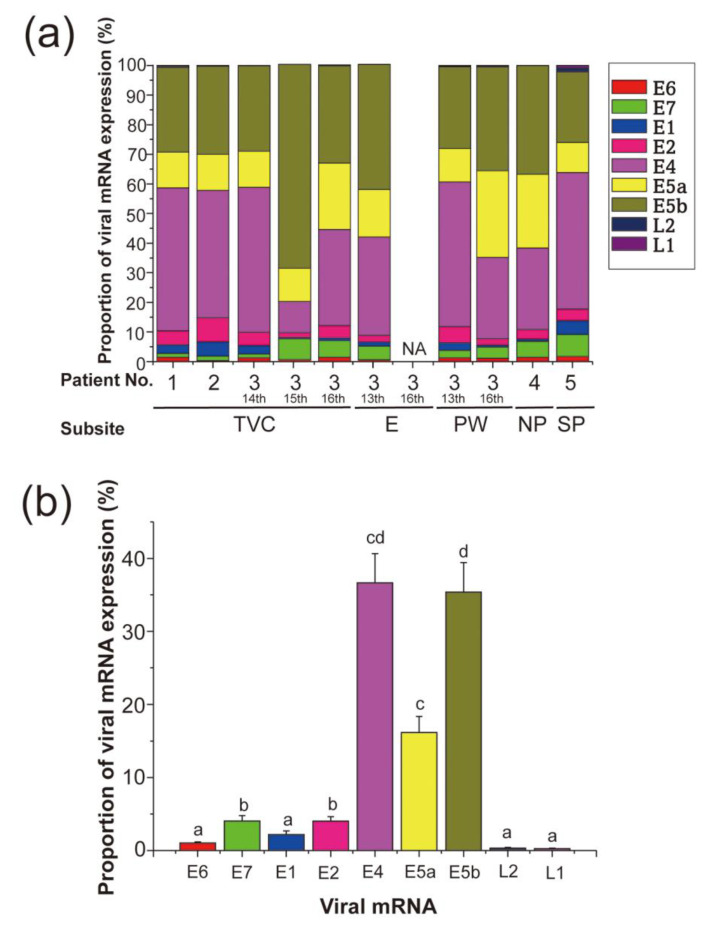
HPV-11 mRNA expression patterns in specimens with HPV-11 infection. (**a**) Viral mRNA expression patterns in individual cases. E, epiglottis; NA, not available; NP, nasopharyngeal papilloma; PW, posterior wall of the hypopharynx; SP, septal exophytic papilloma; TVC, true vocal cord. *E4*, *E5b*, and *E5a* mRNAs were predominantly expressed compared with the other mRNAs in all subsites with HPV-11 infection. (**b**) Proportion of mRNA expression. The probability values (*p*) of the Kruskal–Wallis test were < 0.0001. Values with different characters (a, b, c, d, and cd in Figure 3b) are significantly different among the viral mRNAs (*p* < 0.05, Dwass–Steel–Critchlow–Fligner multiple comparison tests). *E4*, *E5a*, and *E5b* expression together accounted for 88%, while that of *E6*, *E7*, *E1*, *E2*, *L2*, and *L1* was only 12%.

**Figure 4 viruses-13-02024-f004:**
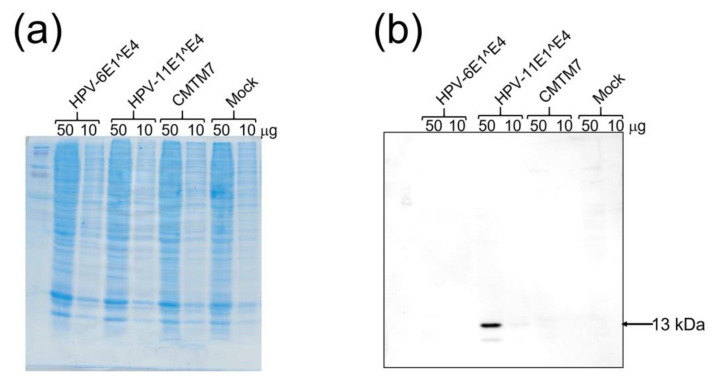
SDS-PAGE of HEK293T cells transfected with various expression vectors and Western blot analysis of the anti-HPV-11 E1^E4 antibody. (**a**) SDS-PAGE analysis with Coomassie brilliant blue staining. (**b**) Western blot analysis of the anti-HPV-11 E1^E4 antibody. Fifty or 10 μg of lysates was loaded from HEK293T cells that were transfected with pcDNA3.1+ HPV-6 E1^E4-3× FLAG, pcDNA3.1+ HPV-11 E1^E4-3× FLAG, pcDNA3.1+ CMTM7-3× FLAG, or pcDNA3.1+ (empty vector). A 13-kDa protein (HPV-11 E1^E4) was observed only in the lane of 50 μg HEK293T lysates transfected with pcDNA3.1+ HPV-11 E1^E4-3× FLAG.

**Figure 5 viruses-13-02024-f005:**
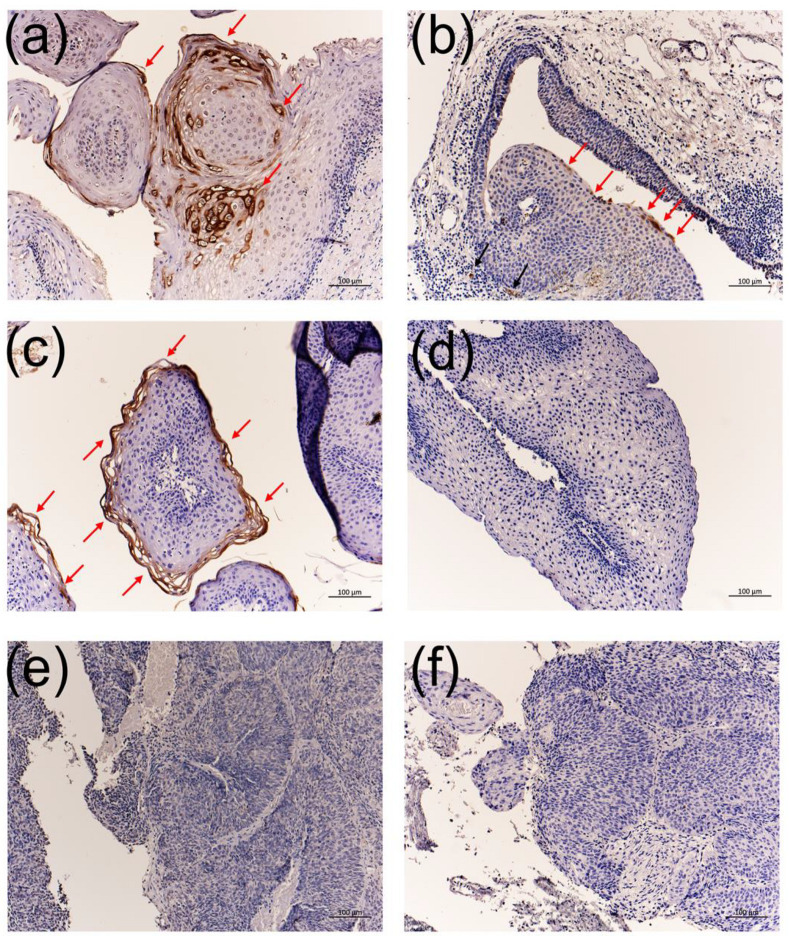
Immunohistochemistry with the anti-HPV-11 E1^E4 antibody. (**a**) HPV-11-infected LP, (**b**) HPV-11-infected nasopharyngeal exophytic papilloma, (**c**) HPV-11-infected septal exophytic papilloma, (**d**) HPV-6-infected LP, (**e**) HPV-16-infected oropharyngeal cancer, and (**f**) HPV-18-infected inverted papilloma with squamous cell carcinoma. Red arrows indicate specific staining. Black arrows indicate a possible non-specific reaction around small vessels. Bars = 100 μm.

**Figure 6 viruses-13-02024-f006:**
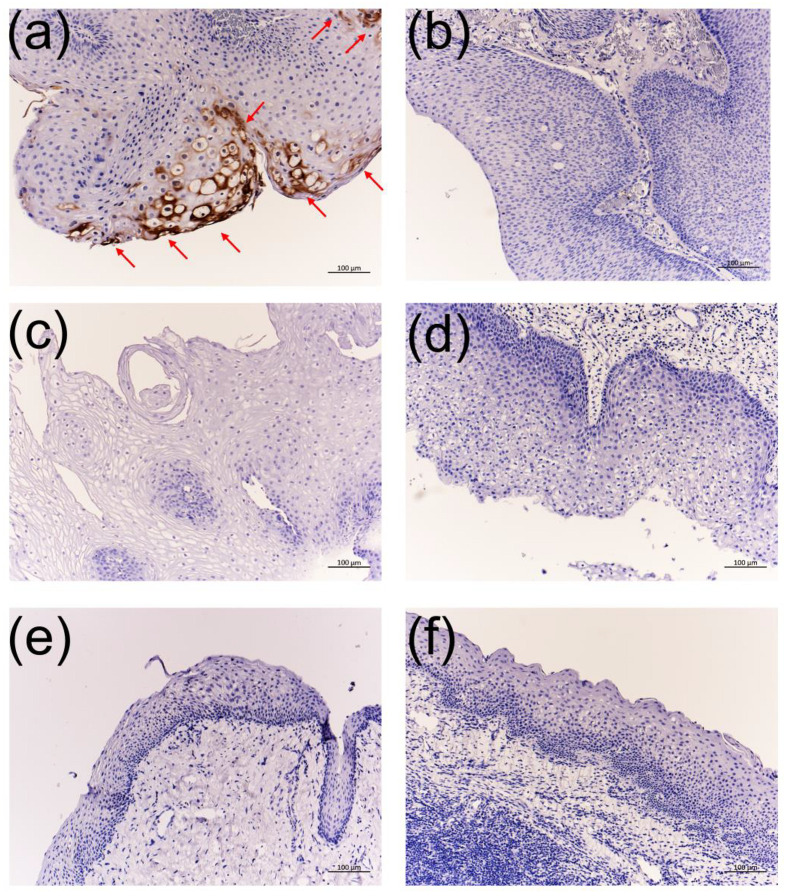
Immunohistochemistry with the anti-HPV-6 E1^E4 antibody. (**a**) HPV-6-infected LP, (**b**) HPV-6-infected nasoseptal exophytic papilloma, (**c**) HPV-negative LP, (**d**) HPV-negative paranasal exophytic papilloma, (**e**) normal vocal cord, and (**f**) chronic tonsillitis. Bars = 100 μm. Nasoseptal exophytic papilloma (**b**) with HPV-6 infection had a significantly lower HPV-6 viral load than the positive cases.

**Table 1 viruses-13-02024-t001:** Immunohistochemistry with anti-HPV-6 and anti-HPV-11 E1^E4 antibodies.

	All Cases (*n* = 66)	Anti-HPV-6 E4 IHC		Anti-HPV-11 E4 IHC	
		Positive	Negative	Positive	Negative
**Laryngeal papilloma**					
HPV-6-positive	20	16	4	0	20
HPV-11-positive	3	0	3	3	0
HPV-negative	5	0	5	0	5
Nasoseptal exophytic papilloma					
HPV-6-positive	1	0	1	0	1
HPV-11-positive	1	0	1	1	0
HPV-negative	2	0	2	0	2
Paranasal exophytic papilloma (HPV-negative)	1	0	1	0	1
Nasopharyngeal papilloma (HPV-11-positive)	1	0	1	1	0
Oropharyngeal papilloma (HPV-negative)	1	0	1	0	1
Inverted papilloma with SCC (HPV-18-positive)	1	0	1	0	1
Normal vocal cord (HPV-negative)	5	0	5	0	5
Chronic tonsillitis (HPV-negative)	5	0	5	0	5
Laryngeal cancer					
High-risk HPV-positive ^1^	5	0	5	0	5
HPV-negative	5	0	5	0	5
Oropharyngeal cancer					
High-risk HPV-infected (HPV-16, 33, 35, 56, 67)	5	0	5	0	5
Tongue cancer (HPV-negative)	5	0	5	0	5

^1^ Determined by high-risk HPV DNA-ISH. IHC, immunohistochemistry; SCC, squamous cell carcinoma.

**Table 2 viruses-13-02024-t002:** Clinical profiles, viral load, and mRNA expression of HPV-11-infected papillomas.

Clinicopathological Characteristics							DNA Viral Load	HPV mRNA Expression/β-Actin								
Case	Age (Years)	Sex	Surgery	Number of Tumors	Subsite	Derkay Score	(Copies/ ng DNA)	*E6*	*E7*	*E1*	*E2*	*E4*	*E5a*	*E5b*	*L2*	*L1*
1	44	M	1st	multiple	Bil TVC	5	64,192	0.11588	0.10541	0.23529	0.39033	3.95554	0.98977	2.33520	0.03630	0.02664
2	44	M	1st	multiple	Bil TVC	7	226,916	0.38194	1.95009	6.36798	10.67637	56.84702	16.06985	39.08463	0.32797	0.11461
3	5	M	13th	multiple	E	5	84	0.00316	0.06096	0.01683	0.02952	0.43326	0.20933	0.54957	0.00101	0.00059
	5			multiple	PW	3	642,111	0.02153	0.04717	0.04567	0.09687	0.87938	0.20339	0.49420	0.00542	0.00482
	5		14th	multiple	Bil TVC	4	230,366	0.03669	0.03956	0.08475	0.13412	1.45394	0.36106	0.85643	0.00581	0.00167
	6		15th	multiple	Bil TVC	5	33,153	0.02311	0.55589	0.03616	0.12762	0.83358	0.89123	5.44958	0.00210	0.00170
	7		16th	multiple	E	2	15,711	NA	NA	NA	NA	NA	NA	NA	NA	NA
	7			multiple	Bil TVC	4	601,070	0.04394	0.20269	0.03255	0.15081	1.16959	0.80580	1.18013	0.00918	0.00594
	7			multiple	PW	2	71,468	0.18674	0.62598	0.11146	0.35295	4.58308	4.86911	5.82166	0.04036	0.06345
4	44	F	1st	multiple	NP	-	11,346	0.17928	0.67253	0.08880	0.41194	3.45012	3.11286	4.59908	0.00536	0.00490
5	63	M	1st	multiple	NS	-	246,077	0.17783	0.77629	0.47749	0.41261	4.80617	1.05967	2.48394	0.13504	0.10001

Bil, bilateral; E, epiglottis; F, female; M, male; NA, not available; NP, nasopharynx; NS, nasal septum; PW, posterior wall of the hypopharynx; TVC, true vocal cord.

**Table 3 viruses-13-02024-t003:** Summary of the relationship between HPV status and immunohistochemical study results.

HPV Status of Specimen	No. of Specimens	Anti-HPV-6 E4 IHC		Anti-HPV-11 E4 IHC	
		Positive	Negative	Positive	Negative
HPV-6-positive	21	16	5	0	21
HPV-11-positive	5	0	5	5	0
High-risk HPV-positive	11	0	11	0	11
HPV-negative	29	0	29	0	29

IHC, immunohistochemistry.

## Data Availability

All data generated or analyzed during this study are included in this published article.

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
