# Peer review of "Development of Antibodies against HPV-6 and HPV-11 for the Study of Laryngeal Papilloma"

_viruses, 2021, doi:10.3390/v13102024_

Round 1

Reviewer 1 Report

Manuscript ID: viruses-1342356

Title: Development of Antibodies Against HPV-6 and HPV-11 for the Study of Laryngeal Papilloma

Authors: Taro Ikegami, Norimoto Kise, Hidetoshi Kinjyo, Shunsuke Kondo, Mikio Suzuki , Narutoshi Tsukahara, Akikazu Murakami, Asanori Kiyuna, Shinya Agena, Katsunori Tanaka, Narumi Hasegawa, Junko Kawakami, Akira Ganaha, Hiroyuki Maeda, Hitoshi Hirakawa

This is an interesting work showing that the developed antibodies against the E4 protein of HPV-6 and HPV-11 can be applied to immunochemistry and further functional studies. This work may be accepted for publication after a careful revision with the following modifications.

Table 2 is unreadable (data are overlapping) and needs to be reorganized.

The results appear to be incomplete. For example, the authors declare that high-risk HPV genotypes were detected in 11/66 (app. 17%) specimens (Table 3). However, there is no other information on these results.

The correlations coefficients between viral DNA load and the levels of the HPV-11 mRNAs should be examined.

They specified that the HPV genotypes determined by PCR were sequenced. Are there any sequence deposits made by the authors? If they have such type of sequence deposit the accession numbers may be provided.

The main additions include a better review of the current literature, especially with regard to the HPV types analyzed, and a more tempered handling of the data and its limitations.  The main limitation of this study is the relatively small number of subjects.

Minor comment:

Lines 43-44 (“Because human papillomavirus (HPV)-6 and 11 infections are usually detected in LP, these infections are thought to be a distinct etiology of LP”): references for this statement should be given.

Lines 333 and 337: The authors use the designation “Figure 2a” and “Figure 2b”, although the photos do not have the designations “a”, “b” … Please check if the numbers of the figures given in the Results section are correct.

Author Response

List of changes and responses to the reviewer’s comments

We thank the reviewer for reading our manuscript and providing valuable comments. We have revised the manuscript in accordance with the reviewer’s comments and suggestions. Our point-by-point responses to the reviewer’s comments are listed below.

  1. The Table 2 is unreadable (data are overlapping) and needs to be reorganized.

Reply

In accordance with your comment, we have reformatted Table 2.

  1. The results appear to be incomplete. For example, authors declare that high-risk HPV genotypes were detected in 11/66 (app. 17%) specimens (Table 3). However, there is no other information on these results.

Reply

The details of high-risk HPV infection was presented in our previous studies. Because the present report focuses on low-risk HPV, we did not include the details of HR-HPV. We have newly cited the two previous studies related to high-risk HPV analysis and added the following sentence to the Materials and Methods section and we have added Table 3, which shows the distribution of high-risk HPV.

Page 3, line 100

The HPV genotypes of the oropharyngeal and laryngeal cancer samples were analyzed in our previous studies [13,14].

  1. The correlations coefficients between viral DNA load and the levels of the HPV-11 mRNAs should be examined.

Reply

In accordance with the reviewer’s comment, we have analyzed the correlation between viral load and mRNA expression levels. Unfortunately, there was no significant correlation between them. Therefore, we have added the following text.

2.9 Statistical analysis

The correlations between viral load and viral mRNA levels were analyzed by Spearman’s rank-order correlation test.

3.2.1 Expression Levels of the Nine Viral mRNAs.

There was no significant correlation between viral DNA load and the mRNA levels of E6, E7, E1, E2, E4, E5a, E5b, L2, and L1.

  1. They specified that the HPV genotypes determined by PCR were sequenced. Is there any sequence deposits made by the authors. If they have such type of sequence deposit the accession numbers may be provided.

Reply

We did not deposit any HPV sequences in the present study because all HPV subtypes we detected have previously been deposited.

  1. The main additions include better review of the current literature, especially with regard to the HPV types analyzed, and a more tempered handling of the data and its limitations. The main limitation of this study is the relatively small number of subjects.

Reply

We agreed with the reviewer’s opinion. However, because the incidence of laryngeal papilloma was reported as 4 per 100,000 children and 2 per 100,000 adults (reference 2, Fortes HR et al), it would be very difficult to obtain a large number of samples with sufficient quality to perform the DNA and RNA analyses.

Minor comment:

  1. Lines 43-44 (“Because human papillomavirus (HPV)-6 and 11 infections are usually detected in LP, these infections are thought to be a distinct etiology of LP”): a references for this statement should be given.

Reply

We added the appropriate reference number.

  1. Lines 333 and 337: The authors use the designation “Figure 2a” and “Figure 2b”, although the photos do not have the designations “a”, “b” … Please check if the numbers of the figures given in the Results section are correct.

Reply

Thank you for pointing out this oversight. Figure 2a and 2b should be Figure 3a and Figure 3b, respectively. We have corrected the numbers accordingly.

Reviewer 2 Report

No comments

Author Response

Thank you for reviewing our manuscript.

Reviewer 3 Report

Ikegami et al have studied the expression of HPV-11 viral gene expression was investigated by quantitative PCR and DNA/RNA in situ hybridization in 66 patients with head and neck lesions, including 28 patients with laryngeal papilloma. In addition, antibodies against the E4 protein of HPV-6 and HPV-11 were evaluated in immunohistochemistry. E4 and E5b mRNAs were expressed from the middle to upper part of the epithelium according to RNA in situ hybridization. E4 immunohistochemistry was expressed in the upper cell layer in accordance with E4 mRNA expression. Other head and neck lesions with HPV-11 infection also showed a positive reaction in E4 immunohistochemistry. The distribution pattern of HPV DNA, viral mRNA, and E4 protein in LP with HPV-11 infection was quite similar to that of HPV-6. In conclusion, these E4-specific antibodies can be applied to further functional studies and clinical applications such as targeted molecular therapies in patients with HPV-6 and HPV-11 infection.

The claims are properly placed in the context of the previous literature. The experimental data support the claims. The manuscript is written clearly enough that most of it is understandable to non-specialists. The authors have provided adequate proof for their claims, without overselling them. The authors have treated the previous literature fairly. The paper offers enough details of methodology so that the experiments could be reproduced.

Minor revisions

Page 2, line 50, add "Human papillomavirus oral papilloma is often sexually transmitted, but non-sexual modes of transmission should be considered (Benyo 2021). "

Page 15, line 452, add "It is expected that prophylactic HPV-vaccines targeting HPV type 6 and 11 will reduce the burden of anogenital warts and laryngeal papilloma (Dunne 2014)."

References

Benyo S, Keane A, Warrick J, Choi KY. HPV-positive oral papillomas in an adolescent-A diagnostic dilemma. Clin Case Rep. 2021 Aug 7;9(8):e04546. doi: 10.1002/ccr3.4546.

https://pubmed.ncbi.nlm.nih.gov/34401152/

Dunne EF, Markowitz LE, Saraiya M, Stokley S, Middleman A, Unger ER, Williams A, Iskander J; Centers for Disease Control and Prevention (CDC). CDC grand rounds: Reducing the burden of HPV-associated cancer and disease. MMWR Morb Mortal Wkly Rep. 2014 Jan 31;63(4):69-72.

https://pubmed.ncbi.nlm.nih.gov/24476977/

Author Response

List of changes and responses to the reviewer's comments

We thank the reviewer for reading our manuscript and providing valuable comments. We have extensively revised the manuscript in accordance with the reviewer’s comments and suggestions, especially the Discussion section.

Our point-by-point responses to the reviewer’s comments are listed below.

Reviewer 1

Page 2, line 50, add "Human papillomavirus oral papilloma is often sexually transmitted, but non-sexual modes of transmission should be considered (Benyo 2021). "
Page 15, line 452, add "It is expected that prophylactic HPV-vaccines targeting HPV type 6 and 11 will reduce the burden of anogenital warts and laryngeal papilloma (Dunne 2014)."

Reply

Thank you for your suggestions.

We have added the recommended text and references.

Reviewer 4 Report

The work of Taro Ikegami et al. substantially uses technologies and experiments already published by the research group in connection with HPV6. In this work, the only difference with what is published is the analysis of a different HPV, the type 11.

I believe that the work needs some improvements in order to be published.

  1. In the introduction, the authors need to explain more clearly why they undertook the analysis of HPV11 after having already published the data for HPV6 in the same type of lesions. In other words, in the current introduction there is only a hint on the possible difference from the clinical point of view of the HPV 11 infection compared to HPV 6 such as to justify this work. This part should be expanded, also using part of the data presented in discussion.
  2. The quantification of the different ORFs in mRNAs has already been published but I believe the authors should explain their data in the light of the polycistronic messengers of HPV.

    Below I show a figure with the possible HPV11 messengers from which it is clear that, for example, E5a and E5b are always present simultaneously in the different mRNAs and therefore the authors should explain how and why they find different levels.

Author Response

List of changes and responses to the reviewer’s comments

We thank the reviewer for reading our manuscript and providing valuable comments. We have revised the manuscript in accordance with the reviewer’s comments and suggestions. Our point-by-point responses to the reviewer’s comments are listed below.

The work of Taro Ikegami et al. substantially uses technologies and experiments already published by the research group in connection with HPV6. In this work, the only difference with what is published is the analysis of a different HPV, the type 11.

I believe that the work needs some improvements in order to be published.

  1. In the introduction, the authors need to explain more clearly why they undertook the analysis of HPV11 after having already published the data for HPV6 in the same type of lesions. In other words, in the current introduction there is only a hint on the possible difference from the clinical point of view of the HPV 11 infection compared to HPV 6 such as to justify this work. This part should be expanded, also using part of the data presented in discussion.

Reply

Thank you for your valuable suggestion. The main purpose of the present study was to create an anti-E4 antibody for HPV-11. From a literature research, we cannot find a good antibody in the immunohistochemistry to detect HPV-11 and HPV-6 without cross-reactivity. Because using PCR to detect HPV-11 and HPV-6 infection is not easy in the clinical setting, we made these antibodies to determine the types of HPV in clinical samples of LP. We have extensively rewritten the Introduction and Discussion to clarify this matter.

  1. The quantification of the different ORFs in mRNAs has already been published but I believe the authors should explain their data in the light of the polycistronic messengers of HPV.

Below I show a figure with the possible HPV11 messengers from which it is clear that, for example, E5a and E5b are always present simultaneously in the different mRNAs and therefore the authors should explain how and why they find different levels.

Reply

Thank you for the your suggestion. We have added text that explains the data comparing the polycistronic messengers of HPV-11 (Pave) in the Discussion section as follows.

According to the PapillomaVirus Episteme (PaVE) database (https://pave.niaid.nih.gov/#explore/transcript_maps/hpv11), the following three promoters have been identified in the HPV-11 genome: E6 promoter (nt 90), E7 promoter (nt 264), and E1 promoter (nts 674-714). The PaVE database also shows 19 polycistronic messages transcribed by these promoters, among which, E5a and E5b are the most encoded genes (n = 18), followed by E4 (n = 15, including the full-length message of E2), E2 (n = 7), E7 (n = 4), E6 (n = 3), L1 (n = 2), and E1 and L2 (n = 1). The expression of each viral gene examined in this study was in good agreement with the number of genes encoded by the message (Figure 3b). According to the PaVE database, E5a and E5b are always transcribed simultaneously, but the level of E5b expression was markedly different from that of E5a, with E5b showing a similar level of expression as E4 in this study. In our previous study, HPV-6 mRNA expression of E5a and E5b differed greatly in LP samples, consistent with the results of the present study. Although the precise reason for the observed differences in E5a and E5b mRNA expression is not clear, the splicing of E5a and E5b mRNAs might occur differently in different LPs. Further study is needed to clarify how mRNA splicing occurs in low-risk HPV.

Round 2

Reviewer 4 Report

The manuscript was substantially improved and is now suitable for publication.